# DoE2Vec: Representation Learning for Exploratory Landscape Analysis

## Abstract

We propose DoE2Vec, a variational autoencoder (VAE)-based methodology to learn optimization landscape characteristics for downstream meta-learning tasks, e.g., automated selection of optimization algorithms. Principally, using large training data sets generated with a random function generator, DoE2Vec self-learns an informative latent representation for any design of experiments (DoE). Unlike the classical exploratory landscape analysis (ELA) method, our approach does not require any feature engineering and is easily applicable for high dimensional search spaces. For validation, we inspect the quality of latent reconstructions and analyze the latent representations using different experiments. The latent representations not only show promising potentials in identifying similar (cheap-to-evaluate) surrogate functions, but also can significantly boost performances when being used complementary to the ELA features in classification tasks.

## 1 Introduction

Solving real-world black-box optimization problems can be extremely complicated, particularly if they are strongly nonlinear and require expensive function evaluations. As suggested by the no free lunch theorem in Wolpert & Macready (1997), there is no such things as a single-best optimization algorithm, that is capable of optimally solving all kind of problems. The task in identifying the most time- and resource-efficient optimization algorithms for each specific problem, also known as the algorithm selection problem (ASP) (see Rice (1976)), is tedious and challenging, even with domain knowledge and experience. In recent years, landscape-aware algorithm selection has gained increasing attention from the research community, where the fitness landscape characteristics are exploited to explain the effectiveness of an algorithm across different problem instances (see van Stein et al. (2013); Simoncini et al. (2018)). Beyond that, it has been shown that landscape characteristics are sufficiently informative in reliably predicting the performance of optimization algorithms, e.g., using Machine Learning approaches (see Bischl et al. (2012); Dréo et al. (2019); Kerschke & Trautmann (2019a); Jankovic & Doerr (2020); Jankovic et al. (2021); Pikalov & Mironovich (2021)). In other words, the expected performance of an optimization algorithm on an unseen problem can be estimated, once the corresponding landscape characteristics have been identified. Interested readers are referred to Muñoz et al. (2015b;a); Kerschke et al. (2019); Kerschke & Trautmann (2019a); Malan (2021).

Exploratory landscape analysis (ELA), for instance, considers six classes of expertly designed features, including $y$-distribution, level set, meta-model, local search, curvature and convexity, to numerically quantify the landscape complexity of an optimization problem, such as multimodality, global structure, separability, plateaus, etc. (see Mersmann et al. (2010; 2011)). Each feature class consists of a set of features, which can be relatively cheaply computed. Other than typical ASP tasks, ELA has shown great potential in a wide variety of applications, such as understanding the underlying landscape of neural architecture search problems in van Stein et al. (2020) and classifying the Black-Box Optimization Benchmarking (BBOB) problems in Renau et al. (2021). Recently, ELA has been applied not only to analyze the landscape characteristics of crash-worthiness optimization problems from automotive industry, but also to identify appropriate cheap-to-evaluate functions as representative of the expensive real-world problems (see Long et al. (2022)). While ELA is well established in capturing the optimization landscape characteristics, we would like to raise our concerns regarding the following aspects.

1. Many of the ELA features are highly correlated and redundant, particularly those within the same feature class (see Škvorc et al. (2020)).

2. Some of the ELA features are insufficiently expressive in distinguishing problem instances (see Renau et al. (2019)).

3. Since ELA features are manually engineered by experts, their feature computation might be biased in capturing certain landscape characteristics (see Seiler et al. (2020)).

4. ELA features are less discriminative for high-dimensional problems (see Muñoz & Smith-Miles (2017)).

Instead of improving the ELA method directly, e.g., searching for more discriminative landscape features, we approach the problems from a different perspective. In this paper, we introduce an automated self-supervised representation learning approach to characterize optimization landscapes by exploiting information in the latent space. Essentially, a deep variational autoencoder (VAE) model is trained to extract an informative feature vector from a design of experiments (DoE), which is essentially a generic low-dimensional representation of the optimization landscape. Thus, the name of our approach: *DoE2Vec*. While the functionality of our approach is fully independent of ELA, experimental results reveal that its performance can be further improved when combined with ELA (and vice versa). To the best of our knowledge, a similar application approach with VAE in learning optimization landscape characteristics is still lacking. Section 2 briefly introduces the state-of-the-art representation learning of optimization landscapes as well as the concepts of (variational) autoencoder. This is followed by the description of our methodology in Section 3. Next, we explain and discuss our experimental results in Section 4. Lastly, conclusions and outlooks are included in Section 5.

## 2 REPRESENTATION OF OPTIMIZATION LANDSCAPE

In the conventional ELA approach, landscape features are computed primarily using a DoE of some samples points $\mathcal{W} = \{w_1, ..., w_n\}$ evaluated on an objective function $f$, i.e., $f \colon \mathbb{R}^d \to \mathbb{R}$, with $w_i \in \mathbb{R}^d$, $n$ represents sample size, and $d$ represents function dimension. The objective function values $f(w_i)$, $i \in \{1, \ldots, n\}$ are the inputs of VAE models in DoE2Vec. In this work, we consider ELA features similar to those in Long et al. (2022), which do not require additional sampling, and compute them with the package `flacco` by Kerschke & Trautmann (2019b;c). These features include commonly used dimensionality reduction approaches such as Principal Component Analysis (PCA) Abdi & Williams (2010), a number of simple surrogate models and many others.

To overcome the drawbacks of the ELA approach, attentions have been focused on developing algorithm selection approaches without requiring landscape features. For example, Prager et al. (2022) proposed two feature-free approaches using a deep learning method, where optimization landscapes can be represented through either 1) image-based fitness maps or 2) graph-like fitness clouds. In the first approach, convolutional neural networks were employed to project data sets into two-dimensional fitness maps, using different dimensionality reduction techniques. In the second approach, data sets were embedded into point clouds using modified point cloud transformers, which can accurately capture the global landscape characteristics. Nonetheless, the fitness map approach suffered from the curse of dimensionality, while the fitness cloud approach was limited to fixed training sample size. Additional relevant works can be found in Alissa et al. (2019); Seiler et al. (2020; 2022); Prager et al. (2021). Unlike these approaches, which were directly used as classifiers, the latent feature sets generated by our proposed approach can be easily combined with other features, such as ELA features, for classification tasks. In our work, we do not propose to replace conventional ELA features, but to actually extend them with autoencoder (AE) based latent-space features. Since the implementation of both approaches mentioned above is not available, a comparison to our work in terms of classifying high-level properties is only feasible by directly comparing their results on a identical experimental setup. Following this, results from the downstream tasks in this work can partially be compared to the mentioned results in Seiler et al. (2022), inlcuding the standard Principal Component Analysis (PCA), reduced Multiple Channel (rMC) and a transformer based approach (Transf.), taking into account that additional hyperparameter tuning was involved in their classification experiments with ELA features.

Our approach is capable of learning the representations of optimization landscapes in an automated, generic and unsupervised manner, with the advantage that the learned features are not biased towards any particular landscape characteristic. Unlike previously mentioned approaches, our proposed method is independent of the sampling method. By using only fully connected (dense) layers that learn from one-dimensional (flattened) landscapes, an AE or a VAE is, in theory, capable of learning any number of input-dimensions without scaling difficulties. Furthermore, the fast-to-train (V)AE models can be easily shared in practice.

## 2.1 Autoencoder

A standard AE usually has a symmetrical architecture, consisting of three components: an encoder, a hidden layer, also known as bottleneck, and a decoder (Figure 1). In short, an encoder projects the input space $\mathcal{X}$ to a representative feature space $\mathcal{H}$, i.e., $e \colon \mathcal{X} \to \mathcal{H}$, while a decoder transforms the feature space back to the input space $d \colon \mathcal{H} \to \hat{\mathcal{X}}$ (see Charte et al. (2020; 2018)). In other words, AE attempts to optimally reconstruct the original input space $\mathcal{X}$, by minimizing the reconstruction error $\mathcal{L}(\mathcal{X}, \hat{\mathcal{X}})$, e.g. mean squared error, during the (unsupervised) training process. Commonly, AE is constructed with an input layer and an output layer of the same dimension $dim(X)$ and a bottleneck layer of lower dimension $dim(H)$, i.e., $dim(H) < dim(X)$, to improve its ability in capturing crucial representations of the input space.

Following this, AE has rapidly gained popularity in the field of dimensionality reduction as well as representation learning Bengio et al. (2013); Tschannen et al. (2018). In comparison with the simple principal component analysis (PCA) technique, AE has the advantage that nonlinear representation features can be learned with activation functions, such as a sigmoid function.

## 2.2 Variational autoencoder

Originating from the same family as traditional AE, a VAE typically has a similar architecture with some modifications, as shown in Figure 1. Unlike AE, the latent space of a VAE is encoded as a

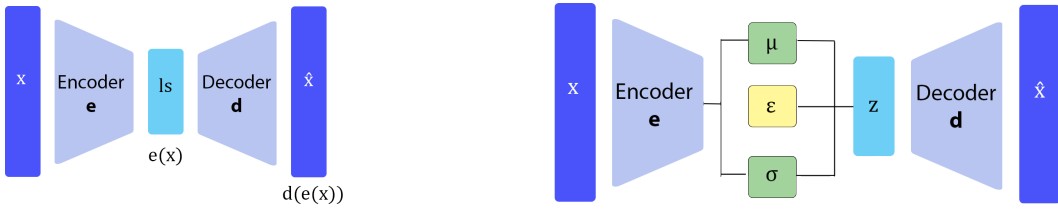

Figure 1: Architecture of a (standard) AE on the left and VAE on the right used in DoE2Vec. The latent space $z$ is defined by $z = \mu + \sigma \cdot \epsilon$, where $\epsilon$ denotes the sampling with mean $\mu$ and variance $\sigma$.

distribution by using a mean and variance layer, together with a sampling method. Following this, the latent space can be properly regularized to provide more meaningful features. During the training process, a common loss function $\mathcal{L}_{\text{VAE}}$ (Equation 1) of VAE is to be minimized, consisting of a regularization term (Equation 2), which can be expressed as the Kullback–Leibler (KL) divergence $\mathcal{L}_{\text{KL}}$, and a reconstruction error, e.g., mean squared error $\mathcal{L}_{\text{MSE}}$ (Equation 3).

$$\mathcal{L}_{\text{VAE}}(\mathcal{X}, \hat{\mathcal{X}}) = \beta \cdot \mathcal{L}_{\text{KL}}(\mathcal{X}, \hat{\mathcal{X}}) + \mathcal{L}_{\text{MSE}}(\mathcal{X}, \hat{\mathcal{X}}), \tag{1}$$

$$\mathcal{L}_{\text{KL}}(\mathcal{X}, \hat{\mathcal{X}}) = \frac{1}{2} \sum_{i=1}^{|\mathcal{X}|} (\exp(\sigma_i) - (1 + \sigma_i) + \mu_i^2), \tag{2}$$

$$\mathcal{L}_{\text{MSE}}(\mathcal{X}, \hat{\mathcal{X}}) = \sum_{x \in \mathcal{X}, \hat{x} \in \hat{\mathcal{X}}} (x - \hat{x})^2, \tag{3}$$

where a weighting factor $\beta$ is included to ensure a good trade-off between $\mathcal{L}_{\text{KL}}$ and $\mathcal{L}_{\text{MSE}}$, $\sigma$ and $\mu$ are the variance and mean latent layers of the VAE and $\hat{\mathcal{X}}$ is the reconstruction of the input space. Detailed explanations regarding VAE can be found in Kingma & Welling (2013; 2019).

### 2.3 BLACK-BOX OPTIMIZATION BENCHMARKING

The development of DoE2Vec is based on the well-known academic BBOB suite by Hansen et al. (2009), consisting of altogther 24 noise-free real-parameter single objective optimization problems of different landscape complexity. For each BBOB problem, the global optimum (within $[-5, 5]^d$) can be randomly shifted, e.g., through translation or rotation, to generate a new problem instance.

## 3 DOE2VEC

Generally, our proposed method uses a VAE with similar design as described in Section 2.2. Precisely, our VAE model has an architecture of altogether seven fully connected layers, where rectified linear unit (ReLU) activation functions are assigned to the hidden layers, while a sigmoid activation function is used for the final output layer of the decoder. The encoder is composed of four fully connected layers with $dim(X)$ depending on the DoE sample size $n$, starting with the input layer size $n$, two hidden layers with sizes $n/2$ and $n/4$ and the latent size $ls$ ($ls < n/4$) for the mean and log variance of the latent space. The decoder is composed of three fully connected layers with sizes $n/4$, $n/2$ and $n$ for the final output layer. The focus of our approach lies on VAE, rather than AE, because it has the additional benefits of regularizing the latent space without loss of generalisation. For comparison, we consider a standard AE model as well (with the same number of hidden layers, except that the latent space is now a single dense layer, instead of a mean and log variance layer with a sampling scheme). Full specifications of the different models are available in our Github repository (for Review (2022)), while pre-trained models are also available on Huggingface.

The general workflow of DoE2Vec can be summarized as follows:

1. First, a DoE of size $2^m$ is created, where $m$ is a user defined parameter. By default, a Sobol sequence by Sobol' (1967) is used as sampling scheme, but any sampling scheme or even a custom DoE can be utilized in practice.

2. The DoE samples, initially within the domain $[0, 1]^d$, can be re-scaled to any desired function boundaries, as the DoE2Vec method is scale-invariant by design.

3. Next, the DoE samples are evaluated for a set of functions randomly generated using the random function generator from Long et al. (2022), which was originally proposed by Tian et al. (2020). The main advantage of using this function generator is that a large training set can be easily created, covering a wide range of function landscape of different complexity.

4. Following this, all objective function values are first re-scaled to $[0, 1]$ and then used as input vectors to train (V)AE models.

5. Lastly, the latent space of the trained V(AE) models can be used as feature vectors for further downstream classification tasks, such as optimization algorithm selection.

In the next section, we will show that the learned feature representations have attractive characteristics and they are indeed useful for downstream classification and meta-learning tasks.

## 4 EXPERIMENTAL RESULTS AND DISCUSSIONS

In this work, we have conducted altogether four experiments for different research purposes. Basically, the first two experiments are to investigate the quality of VAE models trained and to validate the latent space representation learned, while the last two experiments are to evaluate the potential of the DoE2Vec approach for downstream meta-learning tasks, as follows.

1. Analyzing the reconstruction of function landscapes and the impact of the latent size and KL loss weight (weighting factor $\beta$).

2. Investigating the differences in latent space representation between a standard AE and a VAE model.

3. Identifying functions with similar landscapes based on the latent representations.

4. Three downstream multi-class classification experiment to show the potential of the proposed approach in practice, using the latent feature vectors as inputs.

In our experiments, we fix the sampling scheme to a Sobol sequence and $m$ to eight, ending up with a DoE of 256 samples. Unless otherwise specified, all AE and VAE models are trained on a set of $250,000$ five-dimensional ($5d$) random functions.

## 4.1 Reconstruction of Function Landscapes

In our first experiment, we investigate the impact of two model parameters, namely the KL loss weight and the latent size, on the model loss functions. For this purpose, a total of 30 VAE models are trained using combinations of six different latent sizes ($4, 8, 16, 24, 32$) and five different KL loss weights ($0.0001, 0.0002, 0.001, 0.005, 0.01$). Results in the left subplot of Figure 2 clearly show that latent sizes have a positive effect on the $\mathcal{L}_{\text{VAE}}$, with the improvement diminishes beyond 16. On the other hand, the $\mathcal{L}_{\text{MSE}}$ can be improved with smaller KL loss weights, as shown in the right subplot. In Figure 3, the combined effects of these parameters on the $\mathcal{L}_{\text{KL}}$ and $\mathcal{L}_{\text{MSE}}$ are fully visualized.

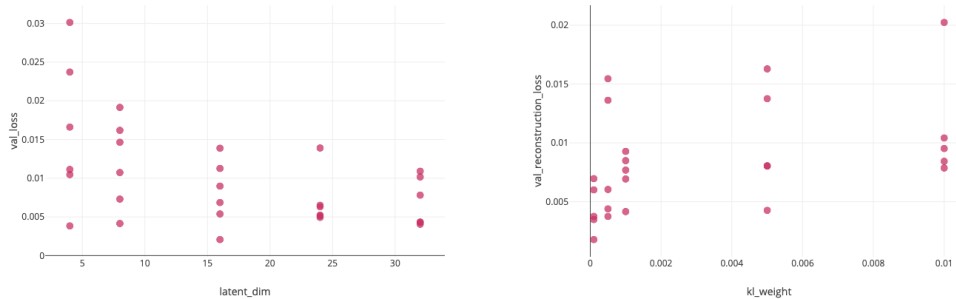

Figure 2: Impact of different latent sizes and KL loss weights on the loss functions. Left: $\mathcal{L}_{\text{VAE}}$ against latent sizes. Right: $\mathcal{L}_{\text{MSE}}$ against KL loss weight.

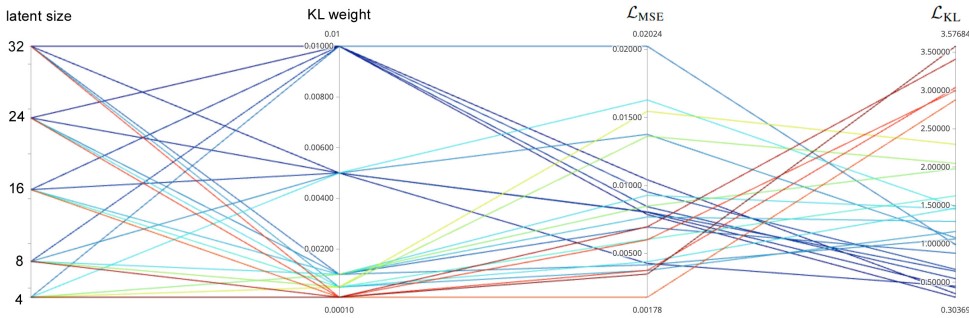

Figure 3: Parallel coordinates plot of different latent sizes and KL loss weights in relation with the validation $\mathcal{L}_{\text{KL}}$ and $\mathcal{L}_{\text{MSE}}$. (Columns left to right: Latent size, KL loss weight, $\mathcal{L}_{\text{MSE}}$ and $\mathcal{L}_{\text{KL}}$.) Each color or line represents a combination of latent size and KL loss weight. Conflict between both loss terms can be observed, since parameter combinations with low $\mathcal{L}_{\text{MSE}}$ have a higher $\mathcal{L}_{\text{KL}}$ and vice versa.

In the remaining experiments, we use a latent size of either $24$ or $32$ (expressed as (V)AE-24 or (V)AE-32) and a KL loss weight of $0.001$ as a good compromise between the two loss terms. It is to be noted that, while the (V)AE architecture can be further improved by applying neural architecture search, we leave it for future work, since the landscape reconstruction is not the ultimate purpose of this work. In fact, the reconstruction here is meant to be a way in evaluating the quality of learned representations. Subsequently, we verify the capability of a VAE-24 model by reconstructing a large variety of functions, using the $24$ BBOB functions (first problem instance) (Figure 4).

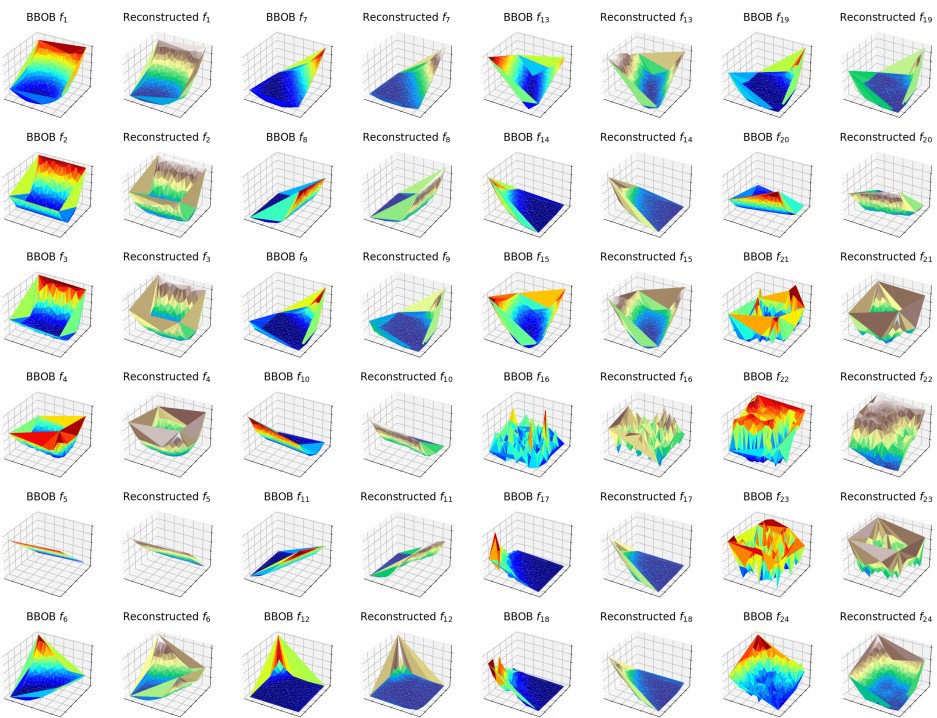

Figure 4: Reconstructions of the 24 $2d$ BBOB funcions (labelled from $f_1$ to $f_{24}$) using a VAE-24 model. The surface plots are generated based on a DOE of 256 samples using triangulation. Generally, the reconstructed landscapes look similar to the actual landscapes based on visual inspection.

## 4.2 LATENT SPACE REPRESENTATION

Next, we project the latent space of an AE and a VAE from our DoE2Vec approach onto a $2d$ visualization using multidimensional scaling (MDS) method, as shown in Figure 5. For a comprehensive comparison, a similar MDS visualization for the feature vectors with all the (considered) ELA features is included as well. Interestingly, the latent space of ELA has a better looking cluster-structure, while the latent space of AE and VAE have a clear structure. As expected, the latent space of AE contains a few outliers and is in general more widespread than the latent space of VAE.

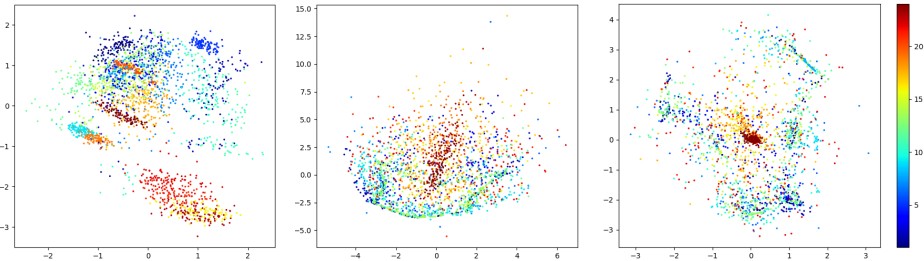

Figure 5: From left to right: $2d$ MDS visualization of the (normalized) ELA feature spaces, the latent space of an AE-32 model and the latent space of a VAE-32 model. Altogether 100 problem instances for all 24 BBOB functions, resulting in a total of 2,400 dots (in each subplot), where each dot represents a feature vector of a BBOB instance and the color denotes the corresponding BBOB function.

To further analyze the differences in latent space representation between the AE and VAE models, we use the generator part of the models to iteratively generate new function landscapes. To be

precise, we select one latent representation as starting point and individually vary each latent variable with small steps between $-1$ and $+1$ of its original value. In Table 1, the generated landscapes are shown only for the extreme cases of an AE-4 and a VAE-4 model. Based on this result, two important conclusions can be drawn.

1. The interpolation for both models is very smooth, showing that the models can learn the landscape representations well and are able to learn the structure, even though the dimensionality of the function landscapes $d$ is not known.

2. The VAE model utilizes a more compact feature representation, where its output variance is greater than those of the AE model for the same variation of latent features.

Table 1: Comparison of function landscapes generated by an AE-4 and a VAE-4 model for different latent variables, using the BBOB $f_{22}$ encoding as starting point $(v_1, v_2, v_3, v_4)$. In each column, the latent features are separately varied by $-1$ or $+1$. Complete variation of the latent features can be found in videos available in our Github repository.

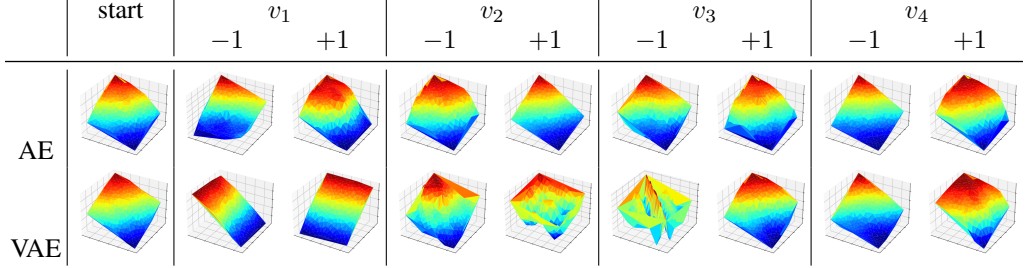

### 4.3 SIMILAR REPRESENTATIVE FUNCTIONS

The first main application of DoE2Vec is to identify cheap-to-evaluate functions with similar latent space representation for a given DoE, which can be very useful when dealing with real-world expensive black box optimization problems, as also described in Long et al. (2022). In this way, large experiments can now be conducted on a "similar" function group at a much lower computational cost. Since a direct analytical verification of the results is very challenging, we instead use a visual inspection based on $2d$ functions to showcase the potential of the proposed method. In Table 2, the 24 BBOB functions are paired with their respective "nearest" random function, where *nearest* is defined as the random function that has the closest feature representation in terms of Euclidean distance. For most BBOB functions, a very well fitting (almost identical) random function can be identified by the DoE2Vec model, where random functions of equal complexity are suggested for the more complex BBOB functions.

### 4.4 CLASSIFICATION TASKS

Secondly, the DoE2Vec approach is designed to learn characteristic representations of an optimization landscape, which can be verified through a classical classification task of high level function properties. These high level properties, such as multimodality, global structure and funnel structure, are important for the ASP, as they often determine the difficulty of an optimization problem. Table 3 illustrates the BBOB functions and their associated high level properties. In this experiment, a standard random forest (RF) model (using `sklearn` from Pedregosa et al. (2011)) is implemented for the multiclass classification tasks based on the latent representations learned by four DoE2Vec models, consisting of AE-24, AE-32, VAE-24 and VAE-32. In other words, the high level properties of a BBOB function are to be predicted using the latent representations. Again, we compare the DoE2Vec approach against the classical ELA method, which is specifically constructed to excel in exactly this kind of function property classification tasks. Beyond that, a combination of classical ELA features with a VAE-32 model is included to evaluate the complimentary effect of the different feature sets.

The classification results (macro F1 scores) of the different feature sets are summarized in Table 4. It is not surprising that the ELA features generally perform very well and outperform the latent

Table 2: Four columns with pairs of BBOB function and their respective nearest random function identified based on latent feature vectors using the VAE-24 model.

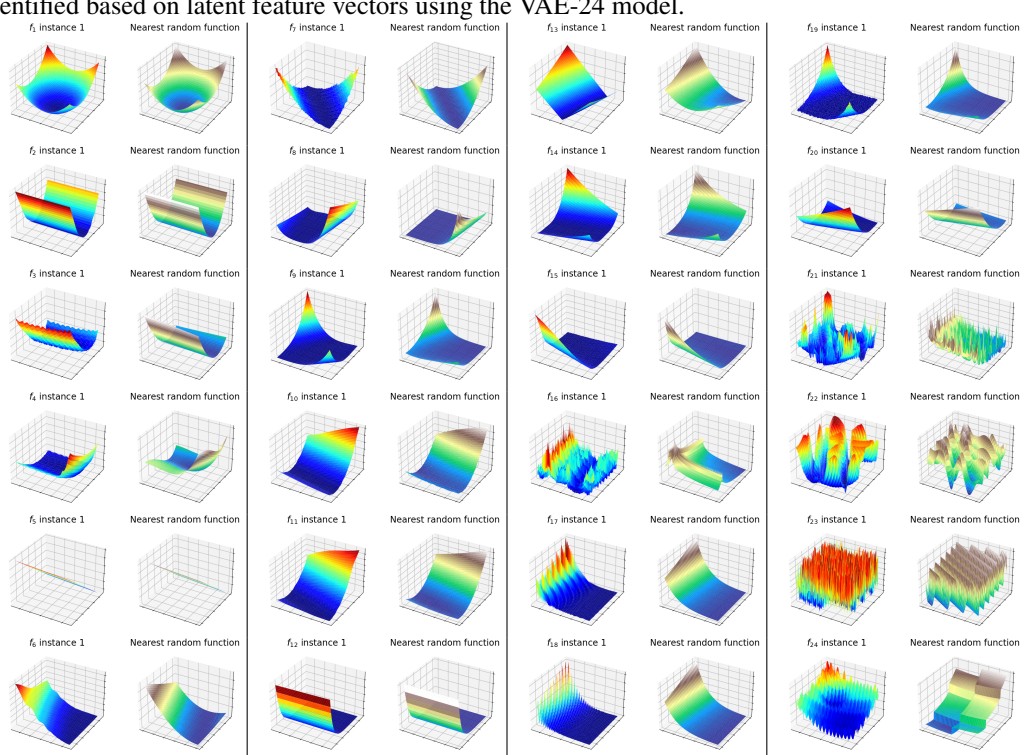

features most of the time, especially in classifying the global structure and multimodal landscapes. Fascinatingly, the classification performances can be significantly improved when the DoE2Vec is combined with the classical ELA method, indicating that both feature sets seem to be complimentary to each other.

## 5 CONCLUSIONS AND FUTURE WORK

In this work we propose *DoE2Vec*, a VAE-based approach to learn the latent representations of an optimization landscape, similar to the classical landscape features. Using a wide set of experiments, we have shown that DoE2Vec is able to reconstruct a large set of functions accurately and that the approach can be used for downstream meta-learning tasks, such as algorithm selection. The proposed methodology can be effectively used next to existing techniques, such as classical ELA features, to further increase the classification accuracy of certain downstream tasks. In fact, DoE2Vec can learn good feature representations for optimization landscapes and has several advantages over the ELA approach, such as feature engineering or selection knowledge is not required, domain knowledge in ELA is not needed and it is applicable to optimization tasks in a very straightforward manner. Nonetheless, there are a few known limitations to the proposed method, such as 1) Our approach is scale-invariant, but not rotation- or translation-invariant. Using a different loss function to train the autoencoders might be able to improve this 2) If a custom DoE sample is used, the model needs to be trained from scratch (no pre-trained model available). This typically takes a few minutes up to an hour, depending on the sample size $n$ and number of random functions to train on, 3) The learned feature representations are a black-box that are hard to interpret directly.

In future work, we plan to improve our approach by tackling some of the challenges mentioned. Apart from that, we would like to verify the usefulness of the random functions with similar latent features w.r.t. the performances of an optimization algorithm.

Table 3: High level properties of the 24 BBOB functions. This table was first introduced in Seiler et al. (2022)

| BBOB function | Multimodal | Global Structure | Funnel |
|---|---|---|---|
| 1: Sphere | none | none | yes |
| 2: Ellipsoidal separable | none | none | yes |
| 3: Rastrigin separable | high | strong | yes |
| 4: Büche-Rastrigin | high | strong | yes |
| 5: Linear Slope | none | none | yes |
| 6: Attractive Sector | none | none | yes |
| 7: Step Ellipsoidal | none | none | yes |
| 8: Rosenbrock | low | none | yes |
| 9: Rosenbrock rotated | low | none | yes |
| 10: Ellipsoidal high conditioned | none | none | yes |
| 11: Discus | none | none | yes |
| 12: Bent Cigar | none | none | yes |
| 13: Sharp Ridge | none | none | yes |
| 14: Different Powers | none | none | yes |
| 15: Rastrigin multimodal | high | strong | yes |
| 16: Weierstrass | high | medium | none |
| 17: Schaffer F7 | high | medium | yes |
| 18: Schaffer F7 moderately ill-cond. | high | medium | yes |
| 19: Griewank-Rosenbrock | high | strong | yes |
| 20: Schwefel | medium | deceptive | yes |
| 21: Gallagher 101 Peaks | medium | none | none |
| 22: Gallagher 21 Peaks | low | none | none |
| 23: Katsuura | high | none | none |
| 24: Lunacek bi-Rastrigin | high | weak | yes |

Table 4: Classification results (averaged macro F1 scores over 10 runs with different random seeds) using a standard RF model with 100 trees, trained on the feature representations (from AE, VAE, classical ELA or ELA combined with VAE-32) of the first 100 instances for each BBOB function and validated on instance 101 to 120. * PCA, rMC and Transformer results are directly taken from the work of Seiler et al. (2022), which uses an identical experimental setup but without repetitions.

| $d$ | Task | AE-24 | AE-32 | VAE-24 | VAE-32 | ELA | PCA* | rMC* | Transf.* | ELA-VAE |
|---|---|---|---|---|---|---|---|---|---|---|
| | multimodal | 0.875 | 0.849 | 0.877 | 0.856 | 0.984 | 0.994 | 0.971 | 0.991 | **0.991** |
| 2 | global struct. | 0.903 | 0.904 | 0.902 | 0.889 | 0.983 | 0.992 | 0.965 | 0.991 | **0.998** |
| | funnel | 0.985 | 0.974 | 0.956 | 0.978 | **1.000** | 0.999 | 0.995 | 1.000 | **1.000** |
| | multimodal | 0.908 | 0.903 | 0.880 | 0.889 | 0.963 | 0.897 | 0.947 | 0.991 | **0.998** |
| 5 | global struct. | 0.838 | 0.828 | 0.810 | 0.793 | **1.000** | 0.807 | 0.859 | 0.978 | **1.000** |
| | funnel | **1.000** | **1.000** | 0.996 | 0.991 | **1.000** | 0.990 | 0.989 | 1.000 | **1.000** |
| | multimodal | 0.877 | 0.813 | 0.844 | 0.838 | **1.000** | 0.839 | 0.952 | 0.974 | **1.000** |
| 10 | global struct. | 0.794 | 0.737 | 0.783 | 0.745 | 0.902 | 0.774 | 0.911 | 0.963 | **0.991** |
| | funnel | 0.998 | 0.993 | 0.997 | 0.993 | 0.972 | 0.977 | 0.991 | **1.000** | 0.997 |
| | multimodal | 0.726 | 0.722 | 0.700 | 0.694 | 0.970 | - | - | - | **0.991** |
| 20 | global struct. | 0.689 | 0.621 | 0.606 | 0.626 | 0.972 | - | - | - | **0.997** |
| | funnel | 0.993 | 0.982 | 0.985 | 0.982 | **1.000** | - | - | - | **1.000** |

# 6 REPRODUCIBILITY STATEMENT

We provide an open-source documented implementation of our package at https://github.com/xxxxxxx/doe2vec, with visualizations and video explanations. Pre-trained models (weights) are available on Huggingface https://huggingface.co/xxxxxx. All models are trained using a single T4 GPU and the computations are carbon neutral ($CO_2$-free) by using solar power. Average training time of a model on $250,000$ random functions was five minutes.

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
