# OpenReview forum: "DoE2Vec: Representation Learning for Exploratory Landscape Analysis"
_ICLR.cc/2023/Conference — Submitted to ICLR 2023_

### Official Review · Reviewer_wn3f · 2022-10-24

**Confidence:** 3
**Correctness:** 2
**Technical Novelty And Significance:** 2
**Empirical Novelty And Significance:** 2
**Recommendation:** 3

**Clarity, Quality, Novelty And Reproducibility:**

The paper is clearly written and the overall structure is good.

The paper simply applies VAEs to vectors of black box functions to retrieve some surrogate function. While for exploratory landscape analysis this might be novel to some degree, the merit of the approach does not get across.

Overall the authors try to give as much details as possible. However, it is not clear in what form the random functions are represented and how they can be accessed in downstream tasks.

**Strength And Weaknesses:**

[+] Designing useful, complementary, and informative features is one of the major challenges in downstream tasks like algorithm selection and therefore reasonable feature-free or automatic feature design approaches are of relevance.

[-] It is claimed that the proposed method facilitates downstream tasks but there are no results contained in the paper to support such claims. More specifically, it is only demonstrated that the method can retrieve random functions which look similar to the actual target functions based on a set of evaluations of the target function in question. While this may indicate that the retrieved random functions may be used as a surrogate it remains open whether (a) the random function is indeed faster to evaluate, (b) the random function is sufficiently close to the target function, and (c) whether the method gives any advantage over standard methods that could be applied directly.

[-] One of the example downstream tasks is algorithm selection. However, it is not clear how the method would be mapped to the setting of algorithm selection where for each problem instance an algorithm needs to be picked. If a problem instance is a bbob function from the paper, one would need to observe objective function values for some algorithms in order to obtain the feature representation. However, if algorithms are already run, the current problem instance is already solved in the best case so that no features are needed anymore. Otherwise, the feature computation would already take too long.

[-] The proposed method is not very innovative. It just applies VAE to some vectors sampled from specific functions. In combination with the limited experimental section regarding the downstream tasks or at least some case studies of how to employ and practically use the approach for downstream tasks, the paper is of little help, I would say.

**Summary Of The Paper:**

In "DoE2Vec: Representation Learning for Exploratory Landscape Analysis" the authors propose a variational auto encoder (VAE) to learn a latent feature representation of black-box optimization problems. In the experimental section, the authors show that this VAE approach is best combined with exporatory landscape analysis (ELA) to obtain the best results. It is argued that the learnt representations can be then used in downstream (meta-learning) tasks such as algorithm selection.

**Summary Of The Review:**

Overall, I think this could be an interesting direction to go for certain BBO tasks but still it is rather unclear to me how the ASP problem should be reasonably implemented with this approach. Also the claims that downstream tasks are facilitated etc. are not proved in this paper. Furthermore, the proposed method is rather an application and more about the dataset used so that the novelty of a methodological perspective is comparably low and questionable to what extent this contribution may impact the community. Therefore, I recommend to reject the paper.

---

> ### Author Response · Authors · 2022-11-07
> **Reply to reviewer wn3f**
>
> Thank you for the review, we have addressed your concerns and answer your questions below:
>
> - It is claimed that the proposed method facilitates downstream tasks but there are no results contained in the paper to support such claims. More specifically, it is only demonstrated that the method can retrieve random functions which look similar to the actual target functions based on a set of evaluations of the target function in question. While this may indicate that the retrieved random functions may be used as a surrogate it remains open whether (a) the random function is indeed faster to evaluate, (b) the random function is sufficiently close to the target function, and (c) whether the method gives any advantage over standard methods that could be applied directly.
>
> Perhaps this was not clear from our wording in the paper but Table 4 shows emperical results on three important down-stream classification tasks that are non-trivial and a main challenge in algorithm selection.
> Apart from this, we kindly refer to the work of Fu-Xing-Long et al. "Learning the characteristics of engineering optimization problems with applications in automo-tive crash" (2022), for more information why finding random surrogate functions is also very useful (but this actually only a side-effect of the proposed approach).
>
> - One of the example downstream tasks is algorithm selection. However, it is not clear how the method would be mapped to the setting of algorithm selection where for each problem instance an algorithm needs to be picked. If a problem instance is a bbob function from the paper, one would need to observe objective function values for some algorithms in order to obtain the feature representation. However, if algorithms are already run, the current problem instance is already solved in the best case so that no features are needed anymore. Otherwise, the feature computation would already take too long.
>
> We understand that the reviewer might not be completely familiar with the field of surrogate-based expensive black-box optimization. In this field it is very common to first take a (relative small) Design Of Experiments on the function to be optimized in order to learn about it's landscape features and also in order to construct a surrogate model that can aid the search and lower the required number of function evaluations significantly. The problem instances we speak of are infeasible to solve using simple gradient descent or exhaustive search and require heuristical search algorithms aided by machine learning methods. It is however too much to explain in this work and we would gladly refer the reviewer to other important works in the field such as "Automated algorithm selection on continuous black-box problems by combining exploratory landscape analysis and machine learning" by Kerschke et al (2019).
>
> - The proposed method is not very innovative. It just applies VAE to some vectors sampled from specific functions. In combination with the limited experimental section regarding the downstream tasks or at least some case studies of how to employ and practically use the approach for downstream tasks, the paper is of little help, I would say.
>
> We have improved the experimental part of the paper to make it more clear what we compare and how. Regarding the VAE component, indeed no obfuscated overly complex method has been used to learn the representations, however the methodology of generating the training set using a random function generator combined with a fixed standardized DOE is novel and AEs are not yet applied to this domain before to solve these problems.

---

### Official Review · Reviewer_KUFH · 2022-10-24

**Confidence:** 3
**Correctness:** 4
**Technical Novelty And Significance:** 3
**Empirical Novelty And Significance:** 3
**Recommendation:** 5

**Clarity, Quality, Novelty And Reproducibility:**


The paper is well-written and well clarified, and original as far as I see.



**Strength And Weaknesses:**


Strength:
1.	The DoE2Vec can reconstruct a large set of functions and can be applied to downstream meta-learning tasks.
2.	The combination of ELA and VAE-32 outperforms all other methods in table 4, which shows the new method can learn a good feature representation for optimization landscapes.
Questions/Weakness :
1.	In Table 2, the VAE-24 model seems cannot represent f16, f21 and f24 well which have medium or high multimodal property. Is there a connection between its bad representation power on these functions with some of the BBOB function properties (such as multimodal level)?
2.	In Table 4, is there other performance measures (except for macro F1 score) can be introduced to quantify the classification results on feature representations from different methods? And it would be more convincing to have multiple executions and report a mean value with variance.
3.	In general do you think your method would increase the computing complexity & time compared with standard ELA for downstream meta-learning tasks?
4.	All in all, I think there are more experiments with repeated runs can be done to make the paper more convincing to the audience.


**Summary Of The Paper:**

This paper proposed a new method to learn optimization landscape features using latent space information with AE/VAE for downstream meta-learning tasks.

**Summary Of The Review:**

All in all, the paper has its novelty and contains a lot of simulations to verify the advantage of their method. However, I would recommend to include more numerical results, such as repeated executions and more performance measures.

---

> ### Author Response · Authors · 2022-11-07
> **Reply to reviewer KUFH**
>
> Thank you for your review and questions. Below our answers to your questions, we also modified the paper according to your suggestions.
>
> 1. In Table 2, the VAE-24 model seems cannot represent f16, f21 and f24 well which have medium or high multimodal property. Is there a connection between its bad representation power on these functions with some of the BBOB function properties (such as multimodal level)?
>
> Table 2 is slightly complicated as we do not see direct reconstructions (those are in Figure 4), but we see randomly generated functions that are most similar in terms of VAE latent representations. For f16, f21 and f24 the similarity of the random functions was actually not that great, and this is due to a limitation in the random function generator we used as it cannot generate such highly multimodal functions very well. Due to space limitations we had left this part of the discussion outside of the paper.
>
>
> 2. In Table 4, is there other performance measures (except for macro F1 score) can be introduced to quantify the classification results on feature representations from different methods?  And it would be more convincing to have multiple executions and report a mean value with variance.
>
> To be able to compare it against recently (2022) published results, of which no source code was available, we used the same experimental setup and performance measure. We did actually perform 10 independent runs with different random seeds and reported the average in Table 4 (standard deviations were marginal). However, we forgot to mention that the reported macro f1 scores are averages and have improved the paper accordingly. We also made the comparison more clear to 4 of the state-of-the-art methods by including them in the table instead of only refering to the results in the original paper.
>
> 3. In general do you think your method would increase the computing complexity & time compared with standard ELA for downstream meta-learning tasks?
>
> Yes, but only slightly. The VAE models are build in under 5 minutes so even if they need to be build from scratch it is very fast. These methods are used in real-world scenarios where the objective to optimize is expensive (in either time or money). Examples are Computational Fluid Dynamics, Ship and car design or automotive crash simulations. For these expensive optimization applications a few additional computation minutes are insignificant.
>
> 4. All in all, I think there are more experiments with repeated runs can be done to make the paper more convincing to the audience.
>
> Thank you for the review and we have adjusted our paper accordingly.

---

### Official Review · Reviewer_7ZSu · 2022-10-24

**Confidence:** 2
**Correctness:** 2
**Technical Novelty And Significance:** 2
**Empirical Novelty And Significance:** 1
**Recommendation:** 3

**Clarity, Quality, Novelty And Reproducibility:**

Clarity:
The paper is well organized.

Quality and Novelty:
The paper does not have any methodological originality and lacks sufficient contribution.

Reproducibility:
The code is not available. I cannot assess the reproducibility.


**Details Of Ethics Concerns:**

no ethics concerns.

**Strength And Weaknesses:**

Strength:

- The proposed VAE-based methodology seems to reconstruct properly most of BBOB functions.

Weaknesses:

- Design of experiments in general is often suited for multivariable analyses and exploring causal relationship. One can test and optimize several variables simultaneously, thus accelerating the process of discovery and optimization. I do not see any analysis which discusses how latent representation of Doc2Vec is suitable for such tasks. It is not clear how each latent vector effectively represents such relationship.

- The reported results are only demonstrating the performance of the latent representation in reconstruction and prediction of properties of a specific set of functions. It seems more like an AE-VAE analysis paper which is evaluating their performances in reconstructing a set of functions.

- I think one important missing aspect is traversal analysis and quantifying the disentanglement of latent factors to show how a disentangled representation can be used to explore the function landscape.

- There is no quantification that shows how Doc2Vec performs against competing algorithms.


Questions / Concerns :

- I do not understand how the proposed approach can be used for downstream meta-learning task?

- What is novel about the results in Figure 2? Adding more dimension to the latent space improves the loss, and a lower beta provides better reconstruction error.  Is that not something already known from $\beta$-VAE?

- Can you elaborate more on the plots in Figure 5?


**Summary Of The Paper:**

The paper proposed a VAE-based approach, called DoE2Vec to learn the latent representation approximating the optimization landscape. The authors showed the latent feature vectors that is learned by the DoE2Vec network can reconstruct and predict high level properties of the function landscapes. They also showed that the proposed method can be used in combination with existing exploratory landscape analysis (ELA), to further improve the classification accuracy of some downstream tasks. The authors reported the effectiveness of DoE2Vec using Black-Box Optimization Benchmark (BBOB) function landscapes.



**Summary Of The Review:**

The authors introduced a VAE-based approach for function landscape analysis which is an interesting and challenging problem. However, I think the authors should justify the role of autoencoders and their latent representation in the ELA task and discuss the role of each latent factor in function landscapes and clarify why this model is expected to outperform the other existing ELA techniques.

---

> ### Author Response · Authors · 2022-11-07
> **Reply to Reviewer 7ZSu**
>
> Thank you for your feedback 7ZSu, much appreciated.
> The design of experiments in our work refers to the initial design of candidate solutions (mostly using a space filling design) for optimizing expensive black box optimization functions (such as computational fluid dynamics).
> We compared the performance of reconstruction but most importantly of classifying high-level optimization landscape features such as multimodality and global structure. These high level features are very important for algorithm selection and automated optimization algorithm selection and tuning. We compared against 5 of the state-of-the-art approaches, however this comparison might have been a bit hidden in the paper due to referencing directly to other work. We modified the paper accordingly.
> In answer to your questions:
> - I do not understand how the proposed approach can be used for downstream meta-learning task?
>
> We demonstrate the use of 3 down-stream classification tasks, namely classifying different optimization landscapes in high-level categories such as highly multimodal functions versus unimodal functions. These tasks are in principle required to perform automatic optimization algorithm selection.
>
> - What is novel about the results in Figure 2? Adding more dimension to the latent space improves the loss, and a lower beta provides better reconstruction error. Is that not something already known from -VAE?
>
> Yes this is partly known, however the effects and magnitude of this trade-off is different per task. We analysed this trade-off to be able to make an informed decision on the hyper-parameters of the proposed model.
>
> - Can you elaborate more on the plots in Figure 5?
>
> In figure 5 one can observe multi-dimensional scaling visualisations of the ELA feature space (a wide variety of 64 computed features for each BBOB function), the AE-32 features, and the VAE-32 features. It is quite clear fromt he ELA feature space that we can already distinquisch separable clusters of BBOB functions (different colors are different function groups). The main difference between AE and VAE here is the regularisation of the feature space, which shows more beneficial behaviour for the VAE case (which we hoped for).
>
> The code is available and can be installed with `pip install doe2vec`, as also stated in the paper.

---

### Official Review · Reviewer_jSUv · 2022-10-26

**Confidence:** 2
**Correctness:** 3
**Technical Novelty And Significance:** 3
**Empirical Novelty And Significance:** 2
**Recommendation:** 3

**Clarity, Quality, Novelty And Reproducibility:**

This paper has good novelty and clarity. However, the theoretical aspects of the method shall be enhanced. The work is reproducible as the required resources to reimplement the work are available online.

**Strength And Weaknesses:**

Strength:

- The paper is well-motivated.
- The application of variational autoencoder in exploratory landscape analysis seems to be novel.
- Four experiments were designed to show the effectiveness of the proposed solution.

Weakness:

- This paper directly applies variational autoencoder to exploratory landscape analysis tasks. There lack more in-depth theoretical analysis of why and how this method works well on these tasks.
- The experiments lack comparison with other approaches.

**Summary Of The Paper:**

This paper introduces a variational autoencoder-based method to learn optimization landscape characteristics for downstream meta-learning tasks. This method uses large training sets to self-learn a latent representation for any design of experiments. Four experiments were designed to evaluate the quality of learned variational autoencoders and their effectiveness on downstream meta-learning tasks.

**Summary Of The Review:**

This paper has clear strengths and weaknesses. Its quality can be improved with more theoretical contribution and more experiments to demonstrate the advantages of the proposed method.

---

> ### Author Response · Authors · 2022-11-07
> **Reply to Official review by Reviewer jSUv**
>
> First of all, thank you for your feedback jSUv.
> Additional in-depth theoretical analysis of the representations is currently ongoing research, in this work we show that
> VAEs can be used to find good feature representations of optimization landscapes which is a start for further research into this direction. Is there any concrete theoretical analysis you would like to see included, we tried to device a wide set of experiments to verify the quality and diversity of the approach with the limited number of pages we can use.
>
> ### Comparison against state of the art
> Our proposed method is actually compared already against *5 state-of-the-art methods*, but perhaps this was not clearly written yet. We compare against ELA (including many feature represenations such as PCA, surrogate models and others), PCA-based, reduced Multiple Channel approach and a Transformer based approach. We compare our proposed approach to the latter 4 directly to the published results of Seiler et al. (2022), due to the methods not being publicly available yet. The Exploratory Landscape Analysis (ELA) feature set is the state-of-the-art approach for representing function landscapes in down-stream classification tasks. We made changes to the paper to clarify this.
>
> We are looking forward to your suggestions.

---

### Decision · Program_Chairs · 2023-01-20

**Decision:**

Reject

**Justification For Why Not Higher Score:**

None of the reviewers argued for acceptance.

**Justification For Why Not Lower Score:**

N/A

**Metareview: Summary, Strengths And Weaknesses:**

This paper explores a methodology based around using a VAE for learning representations of optimization landscapes, with an aim to improve downstream performance e.g. meta-learning of optimization algorithm selection. The paper is interesting and nicely presented, but in its initial form was not sufficiently convincing to reviewers — it was not quite obvious that the experiments in Table 4 of the current draft corresponded to state of the art benchmarks, nor is it quite clear how to interpret the magnitude of these results.

I think this is quite a bit improved from the edits made during the review process (and clarified by the author response), but probably not enough yet to recommend acceptance as-is.

Overall this is a topical paper which will be likely be accepted on resubmission if there is a bit more focus in the presentation on the performance and applicability to downstream tasks (and less on the VAE or AE itself).